# Rat 50 kHz Trill Calls Are Tied to the Expectation of Social Interaction

**DOI:** 10.3390/brainsci11091142

**Published:** 2021-08-28

**Authors:** Candace J. Burke, Mariya Markovina, Sergio M. Pellis, David R. Euston

**Affiliations:** 1Department of Neuroscience, University of Lethbridge, Lethbridge, AB T1K 3M4, Canada; cj.burke@uleth.ca (C.J.B.); pellis@uleth.ca (S.M.P.); 2Department of Psychology, University of Calgary, 2500 University Drive NW, Calgary, AB T2N 1N4, Canada; mariyamarkovina@gmail.com

**Keywords:** ultrasonic vocalizations, communication, play, rats, reward, food

## Abstract

Rats emit a variety of calls in the 40–80 kHz range (50 kHz calls). While these calls are generally associated with positive affect, it is unclear whether certain calls might be used selectively in certain contexts. To examine this, we looked at ultrasonic calls in 30–40 day old male rats during the expectation of either play or food, both of which are reinforcing. Behavior and vocalizations were recorded while rats were in a test chamber awaiting the arrival of a play partner or food over seven days of testing. Control groups were included for the non-specific effects of food deprivation and social isolation. Play reward led to an increase in 50 kHz vocalizations, generally, with specific increases in trill and “trill with jump” calls not seen in other groups. Expectation of food reward did not lead to a significant increase in vocalizations of any type, perhaps due to the young age of our study group. Further, rats that were food deprived for the food expectation study showed markedly lower calls overall and had a different profile of call types compared to rats that were socially isolated. Taken together, the results suggest that trill-associated calls may be used selectively when rats are socially isolated and/or expecting a social encounter.

## 1. Introduction

A predominate theory about the purpose of ultrasonic vocalizations (USVs) in rats is that these calls signal the affective state of the animal [1,2]. Two main categories of calls have been described: 50 kHz calls associated with appetitive situations and positive affect and 22 kHz calls associated with threatening situations and negative affect [2]. While 22 kHz calls are mainly long and flat, 50 kHz calls come in a variety of shapes, including trills, ramps and jumps [3]. Whether the different types of 50 kHz calls have different functional roles is a topic of active research [3,4].

Vocalizations of the 50 kHz type are strongly associated with non-social rewarding stimuli. There is a significant increase in 50 kHz vocalizations emitted when rats are placed in a chamber in which they have received amphetamine (AMPH) [5,6]. Interestingly, the amount of AMPH administered has a direct relationship with the amount of 50 kHz USVs produced [7]. Anticipation of self-administration of electrical stimulation to brain reward centers, such as the ventral tegmental area (VTA) and lateral hypothalamic area, also elicits high rates of 50 kHz calls [8]. The animal in that study showed a marked increase in 50 kHz USVs to cues associated with the electrical stimulation as well as to the stimulation itself. Finally, 50 kHz calls have also been associated with cues indicating food reward [8,9,10] or during anticipation of daily feeding [11]. These findings suggest that 50 kHz USVs signal positive affective states associated with rewarding contexts, independent of social context.

Fifty kHz USVs are also emitted during, and in anticipation of, a variety of rewarding social interactions. Significant increases in 50 kHz calls have been found in males during the anticipatory period before introduction of a female [12]. During copulation, both male and female rats produce 50 kHz vocalizations [13,14]. Interestingly, the number of 50 kHz vocalizations appears to relate to the level of sexual motivation in the respective vocalizing party [12,15]. Juvenile male rats will also emit 50 kHz vocalizations when anticipating the presence of a conspecific, and these vocalizations will increase over days of testing in rats that are socially isolated before testing [16,17]. At least one study has failed to find anticipatory calling in juveniles, but that study used only limited social isolation [18]. Rats will also emit these vocalizations when entering an area frequently visited by other rats [19].

One social context that is known to be particularly rewarding and associated with high numbers of 50 kHz vocalizations is rough and tumble play in juvenile rats. The calls are most common before contact is made [20]. Further, these calls have also been elicited by rats tickled by human handlers, and are more common in isolated than socially housed animals, possibly reflecting the greater value of this hetero-species contact when other social interactions are lacking [14,21]. Rats will also produce 50 kHz vocalizations when introduced to an immobilized and, therefore, easily approachable conspecific and when being introduced to a conspecific after separation [22].

In summary, the 50 kHz USVs are emitted during acquisition and anticipation of non-social and social rewards and also elicit a response from conspecifics. To complicate matters, these calls are also elicited during negative social contexts such as during aggression and when a resident initially meets an intruder [23,24]. Rats also emit 50 kHz calls when a companion is taken away [25]. One explanation for the variety of usage is the 14 potential categories of calls existing in the 50 kHz range [3]. Indeed, the specific calls have been linked to anticipation of play behaviors [16], to mitigate aggression [26], signal play [4], feeding [27] and social contact signaling [25]. Thus, rather than signaling a general positive state, different 50 kHz calls may serve different functional roles.

In this study, we sought to contrast anticipatory calling in juvenile rats to both social and non-social stimuli using play and food, respectively. Two recent studies have attempted similar comparisons. Willey et al. [28] compared vocalizations in male rats to the presence of either food or a female rat on the other side of a wire mesh barrier. The social stimulus elicited far more vocalizations than the food reward. Similarly, Mulvihill and Brudzynski [29] compared vocalizations in males to food reward and to exploration of space recently vacated by an estrous female. The estrous female elicited an increase in 50 kHz calls, especially trill calls, whereas the food reward did not cause an overall change in vocalization rate, but rats did produce more flat calls in the 50 kHz range. This latter finding is consistent with previous reports that feeding is associated with flat calls in the 40 kHz range [27]. These studies show clearly that social stimuli elicit more calls than food reward, but a detailed comparison of calls during *anticipation* of both food and social reward has not yet been reported.

To investigate if anticipation of different types of reward elicited different patterns of calling, we compared the vocalizations of food restricted animals anticipating food to socially isolated animals anticipating play. To ensure that the vocalizations were not due to the restrictions or to the chamber, we had control animals, who were either socially isolated or food restricted, run in the same paradigm as the test subjects but without food or play reward. If a particular 50 kHz call communicates positive affect, we would expect to see elevated rates of this particular call type during anticipation of both food and play. Trill calls are a likely candidate, given their frequency and strong association with drug reward [3]. Any differences in call types or usage, on the other hand, would indicate that 50 kHz vocalizations are more nuanced, signaling specific features of the anticipated reward.

## 2. Materials and Methods

### 2.1. Subjects

Thirty juvenile, male Long Evans, aged 30–40 days, obtained from Charles River (Kingston, NY, USA) at 22 days old were used. These animals were pair housed and given five days to acclimatize to the facility. Eighteen animals were used in the anticipation of play paradigm, 6 in the Play Reward group, who received a play partner after a two minute waiting period, 6 in the Play Control group, who similarly waited for a partner that never came, and 6 as play partners for the Play Reward group. The remaining 12 animals were used for the anticipation of food paradigm, 6 in the Food Reward group, which received food in the test chamber after a two minute waiting period, and 6 in the Food Control group who did not receive food. All animals were maintained on the Lab Diet Enriched Rat Chow (Lab Diet, St. Louis, MO, USA). Housing rooms were lit during the day and dark at night and all testing occurred during the day.

### 2.2. Behavioral Procedure

The testing enclosure was a Plexiglas box (50 × 50 × 50 cm), which was situated inside a soundproof chamber (61 × 61 × 83 cm) lined with acoustic foam. The floor of the chamber was covered with 2 cm of paper-based bedding (Care Fresh, Ferndale, WA, USA) which we found to facilitate play while causing very low levels of ultrasonic interference. Ultrasonic vocalizations were collected using a specialized microphone (Model 4939, Brüel & Kjaer, Denmark) with a frequency response of 4 Hz to 100 kHz. The microphone was located in the ceiling of the chamber and was approximately 15 cm above the center of the play enclosure. The microphone was connected to a Soundconnect™ amplifier (Listen, Inc., Boston, MA, USA) and sound waves were recorded at 195,313 Hz using 16-bit resolution via a multifunction processor (model RX6, Tucker-Davis Technologies, Alachua, FL, USA). Video was recorded using a USB webcam (Microsoft Lifecam Studio, Redmond, WA, USA) with its infrared filter removed, positioned directly above the animal

### 2.3. Anticipation of Play Test

Data presented were taken from a 2 min anticipation period during which a target animal either waited in the testing enclosure for the arrival of a familiar play partner (i.e., his former cage mate) or received no partner. For the Play Reward group, once the play partner was introduced, animals were allowed to play for 10 min, following previously established methods [30]. After testing, rats were returned to their original home cages for an additional hour of play and then separated. The Play Control animals, who received no partner, waited in the chamber for 10 min, and then were placed back in their home cage. One hour later, these animals were introduced to their former cage mate for 1 h and 12 min of play before separation. Prior to testing, animals were individually habituated to the enclosure for 10 min each day for 3 consecutive days. On the 3rd day all subjects were socially isolated from their cage mates for 24 h prior to play testing and isolation continued until after all 7 days of testing were complete, in order to increase overall playfulness [31,32,33]. Both habituation and testing sessions were conducted in complete darkness, as this has been shown to facilitate USV production [17]. Audio and video recordings began after the target rat was placed in the test enclosure. Because both audio and video data were recorded on separate devices, a custom-made beeper with an LED light was used to emit a simultaneous light/sound cue at the beginning and end of each recording session and these times were used to synchronize audio and video recordings during subsequent analysis. Following each session, the apparatus was thoroughly cleaned with Virkon, a broad-spectrum disinfectant (Virkon Disinfectant Technologies, Sudbury, United Kingdom), and bedding was replaced to avoid any odors from other subjects. The data analyzed comes from day 1 and day 7 in all animals with the exception of one animal in the Play group who was not separated from his cage mate after testing on day 6. For this one animal, we use data from day 6 instead.

### 2.4. Food Restriction

In order to food restrict animals at such a young age, we used the play animals as weight controls. Each food-restricted animal was matched based on weight to a play animal when handling started. The target weight was calculated based on that of the play animal. The food restriction animals were restricted to maintain 85% of the weight of play controls. The animals were separated for three hours to eat the appropriate amount of food and then were placed back in with their cage mate, with any food remaining in their isolated chambers being placed in with both the animals.

### 2.5. Anticipation of Food Test

The Food Reward group consisted of 6 animals who anticipated food reward in the chamber for 2 min and subsequently received half a semi-sweet chocolate chip each 30 s for 10 min. The chocolate chips were dropped by the experimenter from the top of the sound chamber. The animals were then brought back to their individual feeding cages, which had their allocated food, and then were given 3 h to eat before being return to their shared cages. The remaining 6 animals, the Food Control group, were placed in the chamber for 12 min while the experimenter remained in the room; however, no food was given. These animals were similarly isolated and given 3 h to eat before being return to their shared cages. One hour into this period, the Food Control rats were given 10 chocolate chips so as to equate both the quantity and type of food eaten each day between the Food Reward and Food Control groups.

### 2.6. Ethics

All procedures were in accordance with the University of Lethbridge institutional animal care and use committee and Canadian Council on Animal Care recommendations and guidelines.

### 2.7. Behavioral Analyses

The 2 min anticipatory period was analyzed in each group. The behaviors were coded using recorded video sequences and were evaluated at normal speed, slow motion and frame-by-frame to manually code these behaviors [23,34]. To capture the range of possible actions, behavior patterns associated with anticipation were scored (Table 1).

Both the type of behavior and duration of that particular behavior were scored manually. Importantly, we assigned a behavioral category at every video frame, so that no time was left unaccounted for. This meant that the video frame of the termination of each behavior was the beginning of the next behavior. Frame-by-frame analysis of video was performed using Avidmux software, and the behaviors scored are shown in Table 1.

### 2.8. Vocalization Analyses

Acoustic data were analyzed using Raven Pro 1.4 software (Bioacoustics Research Program, Cornell Lab of Ornithology, Ithaca, NY, USA). The Raven Pro software generated spectrograms with a 256-sample Hann window from which the experimenter manually selected 50 kHz vocalizations. The 14 different 50 kHz vocalizations characterized by Wright et al. [3] were scored as distinct calls as we have done previously [16,26]. The occurrence of these calls was used to compare rates of calling, types of calling and whether different types of calls were associated with particular types of actions.

To compute the proportion of each call category emitted by each group, we summed the total number of each call for that group (e.g., total number of trills across all six Food Reward rats) and divided by the total number of all calls emitted by that group and expressed the result as a percentage. Analysis was based on the entire 2 min anticipation period. We also analyzed the change in vocalization rate from day 1 to day 7 for each call category. For this analysis, we first computed the rate of calling on day 1 and day 7 for each rat for each call, and then expressed this as a difference score (i.e., day 7 rate—day 1 rate). Difference scores were then averaged to compare the change in vocalization rate for each call category for each group. A similar method was used to analyze the vocalization rates for each call when the two play and two food groups were combined, except that all data was from day 1 and results are shown as raw vocalization rates.

### 2.9. Statistical Analyses

To evaluate the associations between all the behaviors and call types a Monte Carlo Shuffling method was used [16]. We first counted the number of co-occurrences of each vocalization type with each of the coded behavioral categories. A vocalization was counted as occurring during a particular behavior if the mid-point of the call occurred between the start and stop time of the behavior. To allow for small errors in coding of the start and stop times of behaviors, the window for counting a call as associated with a behavior was extended to 200 milliseconds before the start of the behavior and 200 milliseconds after the end of the behavior. Shuffling was achieved by assigning each vocalization a random time within the duration of the 2 min observation period. Hence, the relative frequency of vocalizations was kept the same for each shuffle. This shuffling was done 10,000 times and the total number of co-occurrences of each vocalization type with each type of behavior was tabulated. Based on the distribution of these counts, a z-score was calculated for each of the actual co-occurrences of each call–behavior pairing. The higher the z-score, the more likely that specific combination of call and behavior could have occurred by chance (i.e., for *p* ≤ 0.05 the z score is +1.96 and for *p* ≤ 0.01 the z score is +2.58). Large negative z-scores, on the other hand, indicate that the call and behavior are associated much less than expected by chance. Shuffling was performed separately for each animal and the z-scores averaged across animals in the same group to generate the final, average z- score values.

## 3. Results

### 3.1. Vocalizations

#### 3.1.1. Vocalization Counts

To gage anticipation, we calculated the average vocalizations produced during the two minute period of anticipation for each group. As is evident in Figure 1, by day 7 of testing the Play Reward group had a significantly greater average vocal production than the other conditions. A repeated measures ANOVA was conducted on the influence of group (Play Reward, Play Control, Food Reward and Food Control) on the vocalizations produced on day 1 and 7 of testing. The effect for testing day was not significant *F*(1, 4) = 0.233, *p* = 0.654, partial η^2^ = 0.055, but the group *F*(3, 12) = 9.10, *p* = 0.002, partial η^2^ = 0.695 and testing day X group interaction were significant *F*(3, 12) = 5.49, *p* = 0.013, partial η^2^ = 0.578. The Play Reward group increased vocalizations production in the anticipatory period, explaining the significant effect of group, however, overall the other groups did not show an increase; in fact, the two control groups actually decreased over days.

#### 3.1.2. Vocalization Analyzed by Category

To assess if anticipation of different rewards impacted the type of vocalizations produced, we calculated the average number of each call subtype emitted over the entire 2 min anticipation period in each condition and expressed this as a proportion of all calls (Play Reward, Play Control, Food Reward, Food Control). To assess if the calls emitted changed over days of testing we performed this analysis on both day 1 of testing, when the animals had been habituated to the chamber but had not experienced reward, and day 7 when the reward groups had received 7 days of experience with rewards and the chamber and the control groups had 7 days experience with the test chamber. The analysis, shown in Figure 2, reveals several interesting patterns. First, both food deprivation and social deprivation appear to influence the types of calls produced on both days. Secondly, the pattern of calling on day 1 is similar for the Play Reward and Play Control groups, as is the pattern of calling in the Food Reward and Food Control groups, but the play and food conditions differ markedly. In particular, the rats in the food conditions exhibited a much higher proportion of flat and upramp calls and proportionally fewer trills than the rats in the play conditions. Thirdly, while the rats in the play groups mostly had minor changes in call distribution from day 1 to day 7, the rats in both food groups had a large increase in trill and trill with jumps and a reduction in flat calls. In fact, by day 7, the majority of calls from these rats were trills, trills with jumps and upramps. The rats in the play conditions, in contrast, had a wider variety of call types on day 7. This reduction in the variety of calls in the two food conditions on day 7 was also validated by a comparison of Gini coefficients [35]. On day 1 in the Play Reward and Play Control conditions, the Gini coefficients were 0.60 and 0.58, respectively, while on day 7, they were barely different at 0.58 and 0.54. In contrast, in the Food Reward and Food Control conditions on day 1, the Gini coefficients were 0.54 and 0.58, respectively, but then increased to 0.70 and 0.79.

To quantify these effects, we also compared the change in the average number of vocalizations of each type from day 1 to day 7, computing a change score for each vocalization. In Figure 3A, it is apparent that the Play Reward group showed increases in trills, trills with jumps and upramps. A two-tailed *t*-test was used to compare these change scores for control and reward groups for each vocal category. Compared to the Play Control group, the increase in calls was significant for both the trill (*t*(10) = 2.81, *p* = 0.019) and trill with jump (*t*(10) = 2.70, *p* = 0.022) calls. In contrast, Figure 3B shows that the Food Reward group showed an increase in trills and a decrease in upramps and flats, but none of these were statistically different compared to the changes in the Food Control group. Hence, the anticipation of social reward seems to lead to an increase in calls with trills (trills and trills with jumps), whereas the anticipation of food does not cause unique changes in the number of any types of calls.

As previously mentioned, our qualitative analysis (Figure 2) revealed dramatic differences between play and food groups in the types of calls used on day 1. To examine this effect in more detail, we combined the day 1 data from the Food Reward and Food Control groups and separately combined the data from the Play Reward and Play Control groups. As none of these groups had yet to experience the associated reward, there is no reason to suspect differences within the Play Reward/Play Control or Food Reward/Food Control supergroups. Hence, the only difference between the play and food supergroups is that one was socially isolated (play groups) and the other food deprived (food groups). Figure 4 shows a comparison of the average number of calls emitted by each group during the anticipation period, broken down by category. A two-tailed *t*-test was used to compute the probability of a difference between the play groups and food groups for each call category. Both groups emitted more trills than any other calls, but the play groups emitted far more trills than the food groups (*t*(22) = 3.10, *p* = 0.005). Significant differences were also seen between play and food groups in trills with jumps (*t*(22) = 2.39, *p* = 0.026) and short calls (*t*(22) = 3.08, *p* = 0.005) although the strength of this latter effect is due to the fact that there were zero short calls emitted by the food deprived animals on day 1. In sum, when placed in a new environment, rats that have been socially isolated emitted more trill, trill with jumps and short calls compared to rats that had been food deprived but not socially isolated.

### 3.2. Behavior

We compared the mean time spent in each of the coded behaviors on day 1 and day 7 for each of the four treatment groups. We then grouped these measurements into slow locomotion (step, turn or walk) and fast locomotion (run or jump). The latter is of particular relevance because it could indicate the level of arousal. As shown in Figure 5, the two food groups showed no change in the average time spent in slow locomotion from day 1 to day 7, while both play groups showed a slight decrease. A two-way ANOVA with between-subjects factor group (Play Reward, Play Control, Food Reward, Food Control) and repeated-subjects factor day (1 or 7) showed only an effect of group (F(3, 20) = 3.64, *p* = 0.03, partial η^2^ = 0.35). Tukey’s HSD tests for multiple comparisons showed that the primary reason for this group effect was a significant difference between the Food Control and Play Control groups (*p* = 0.046). With fast locomotion, all groups showed an upward trend from day 1 to day 7 and this was borne out in a two-way ANOVA (group x day), which showed a significant effect of day (F(1,20) = 24.20, *p* < 0.001, partial η^2^ = 0.548). There were also differences between groups (F(3,20) = 4.458, *p* = 0.015, partial η^2^ = 0.401), which Tukey’s HSD tests revealed were primarily due to a significant difference between the Food Control and Play Control groups (*p* = 0.026). Hence, we see effects of group on both slow and fast locomotion with play groups showing less slow locomotion and more fast locomotion but no significant differences in the rate of change of either variable from day 1 to day 7.

Two other behaviors, rearing and exploring, stood out because they apparently showed different patterns for the Play Reward group compared to the other groups. For exploration, the Play Reward group showed a reduction in duration from day 1 to day 7, but a two-way ANOVA (group × day) showed no significant effects of group, day or their interaction. Similarly, rearing duration increased from day 1 to day 7 only in the Play Reward group, but a two-way ANOVA (group × day) failed to show any significant differences, although the day X group interaction was close, with a *p*-value of 0.055.

### 3.3. Vocal-Behavioral Associations

The vocal-behavior correlations shown in Figure 6 demonstrate several differences both between groups and days tested. First, on day 1, the Play Reward and Play Control groups had very similar profiles, with the strongest associations being between trill calls and walking, and between downramp and running. When comparing between groups on day 7, the Play Reward and Play Control groups were less similar. For the Play Reward group, the majority of the strong behavior-call associations involving running and jumping, whereas for the Play Control group, the majority of strong associations were with walking and running. More specifically, in the Play Reward group there were strong associations of downramp, flat and split with jumps. Trills, flat/trill combinations and trills with jumps were associated with running. In the Play Control group, on the other hand, the strongest associations were between trills and walking, and between downramps and running. On day 1, the Food Reward and Food Control also had similar profiles, with the trill-walk, upramp-walk and flat-walk being the predominant associations. Arguably the most interesting finding is that by day 7, both the Food Reward and Food Control groups did not have any significant vocal-behavior associations.

## 4. Discussion

The primary goal of this study was to compare the vocalizations emitted by male rats in anticipation of two types of reward: food and social play. Care was taken to equate the age of the animals being tested, as social behavior and vocalizations change dramatically with age [36]. We also included controls for the effects of social isolation and food deprivation, as both might be expected to affect the production of vocalizations irrespective of the presence of rewards. Over seven days, repeated pairing of the recording chamber with the reward of a play partner led to an increase in 50 kHz vocalizations, a change not present in the social isolation control group. There were also trends towards greater high-intensity movement, less exploration and increased rearing in the group rewarded with play, although none of these behavioral changes were statistically significant. Examining each call category individually showed that both the trill and trill with jump calls increased with increased training, suggesting that these two calls in particular may have a social role. In contrast, repeated pairing of the reward chamber with food did not lead to any discernable changes in either vocalizations or behavior.

A secondary finding was the robust, but different effect of social isolation and food deprivation on vocalizations. Hungry rats produced fewer and different calls than socially isolated ones. The lower number of calls in the food-deprived animals is apparent in Figure 1 and this finding is consistent with previous reports that food-deprived animals call less [37,38]. Differences in the distribution of calls is apparent in Figure 2 on both days 1 and 7, where distributions look similar within the two play conditions and within the two food conditions, but very different between the food and play super-groups. A quantitative comparison of call rates on the first day of testing showed that the primary difference between play and food groups was in trills and trills with jumps, the same two calls whose prevalence correlates with the expectation of social reward. This adds to the evidence that these calls have a social role. The increased drive for play induced by social isolation increases their prevalence and the expectation of the arrival of a play partner increases their prevalence further.

Our finding that trills and trills with jumps are tied to the expectation of social reward is broadly consistent with the findings of others. Earlier studies have shown that the expectation of social play in juveniles increases the emission of 50 kHz calls, generally [17]. While this early study did not categorize calls, a later study showed that frequency-modulated 50 kHz calls predominate during play itself, at least among juvenile males [14]. Our results are also similar to the pattern reported by Wright et al. [3] who found that two male adult rats placed in a chamber together after saline injection emitted, as the most frequent call categories, 20% trills, 17% flat/trills and 20% flats. The increased frequency of flat-containing calls in that study may be either because of recent injection or because adult males tend to have more aggressive encounters, which have been linked to 50 kHz flat calls [14]. More recently, Mulvihill and Brudzynski [29] showed that trill calls, in particular, are more common as male rats explore a space recently vacated by an estrous female. Mating is also commonly associated with 50 kHz calls in the period before ejaculation and 22 kHz calls afterwards [39]. It would be interesting to compare the types of 50 kHz calls emitted during sex with those during play and other affiliative behaviors, but most studies of sexual vocalizations were conducted well before the common use spectroscopic analysis. One relatively recent report shows that the calls during sex are frequency modulated, but does not categorize calls any further [14]. In studies very similar to the present one, we have previously found high rates of both trills and trills with jumps in male rats anticipating social reward (but there we did not demonstrate that rates were modulated by social expectation) [16,40]. Further, when male juvenile rats play, the most commonly emitted calls are trills and trills with jumps [4]. Taken together, the data suggest that trills and possibly trills with jumps play a role in calling to other rats, possibly to broadcast a general state of positive affect and/or to attract them [1,2,41,42]. Trills, in particular, are the most common call detected in many studies, suggesting that, although trill rate is modulated by social expectation, rats may be set, by default, for constant social signaling [3,16,43].

The lack of anticipatory vocalizations or behavior in our food condition is puzzling, especially in light of several other studies showing an increase in 50 kHz calls during the expectation of food. Willey and Spear [28] showed elevated 50 kHz calling when male rats were placed in a chamber with food on the other side of a barrier. Rats also show an increase in 50 kHz calls in the 15 min before their daily feeding [11]. Several studies have shown that 50 kHz vocalizations increase after presentation of a tone or light cue that predicts food delivery [8,9,10,38]. The one exception to this pattern was a study by Tripi et al. [44], which showed that Pavlovian conditioning with lever, light and food did not lead to elevated cue-related calling, though the anticipatory period was relatively short (8 s). The weight of the evidence, however, suggests that rats will elevate their calling when context or cues predict the arrival of food.

Why did others find increases in 50 kHz vocalizations associated with food expectation while we did not? The contrast is most striking with Burgdorf et al. [8], upon which our study was modelled. Both studies used the same strain of rat, the same 2 min expectation window, and similar methods of food deprivation. Burgdorf et al. did use a mix of male and female rats while we used solely males, but previous studies have shown that sex differences in vocalizations are either subtle or non-existent [45,46,47] (see below). Hence, the most notable difference was that we used juvenile rats while they, and all the other studies cited above, used adults. The idea that anticipatory vocalizations for food are age-specific is consistent with a previous report that male adolescent rats show lower levels of food-associated vocalizations than adults [28].

Another explanation for our lack of food expectancy calls is that our rats were not as motivated by food. The use of juveniles was necessary because we wanted to compare the effects of food and play in similar groups of animals and juveniles exhibit a pronounced peak in play activity between 30–40 days of age [33,48]. Unfortunately, this imposed constrains on our ability to food-deprive our animals, because prolonged caloric restriction at this age leads to stunted growth. In our study, a control group given free access to food (in this case, the two play groups) were used as a control to set weight targets for rats as they grew. However, as the food-deprived rats probably did slow their growth, the freely fed rats may have served as an overly generous target for our 15% weight reduction. As a consequence, the motivation to seek out food reward may have been reduced in our food groups, resulting in the lack of anticipatory behavior for food reward. On the other hand, we used a highly palatable food reward (chocolate chips) that at least one other study showed was sufficient to induce 50 kHz vocalizations even in rats that were not food deprived [28].

A third possible explanation for our lack of food expectancy calls is that we simply lacked the statistical power to detect increased vocalizations for food reward. There are intriguing non-significant differences in the change in vocalizations from day 1 to day 7 between the Food Reward and Food Control groups (Figure 3B) that suggest that a larger number of subjects might have allowed detection of some differences. On the other hand, even with the low number of animals, there were very clear differences in the Play Reward and Play Control conditions (Figure 3A), suggesting that low power was not a critical limitation. To us, the most likely explanation is simply that rats at this age, due to some biological programming, simply care far more about social activities than they do about food.

While our study did not allow us to determine which calls are tied to food reward, there is considerable evidence from other studies. Many studies use only broad categories of calls. For example, Opiol et al. [11], showed an increase in frequency modulated calls tied to a tone that predicted the delivery of food. Other studies are more specific. In one, male rats were trained for 24 days to expect food after a light cue [9]. In the 2 min anticipatory window, the calls related to the expectation of food were “other frequency modulated” (which excludes calls with trills), “step frequency modulated” (which look like Wright et al.’s category, split) and 50 kHz flat calls. Similarly, Brenes and Schwarting [10] found an increase in step calls over flat and trill calls during cued anticipation of food reward. In a recent study that directly examined the different calls elicited by food and social stimuli, male rats were allowed to explore a space with either a highly palatable food reward or an empty space recently occupied by an estrous female [29]. Flat calls were more common in the food group while calls with trills were more common in response to the female. The elevated flat calls are consistent with previous reports of 40 kHz flat calls related to food consumption [11,27]. Taken together, the evidence suggests that the expectation of food elicits flat, step, split and other frequency-modulated calls but notably, not calls with trills. This is a striking contrast with the social-related calls we observed, both of which include trills.

Further insights into the function of different 50 kHz calls can be gleaned from studies with amphetamine, a highly reinforcing drug. Amphetamine induces a robust increase in 50 kHz vocalizations both from acute administration or to contextual cues associated with the drug [5,49,50,51]. Some studies have found that amphetamine increases all types of 50 kHz calls [52]. Another study by the same first author found increases in flat, trill, complex, inverted U, short, step up, multistep, upward ramp calls [43]. Other studies have found more selective effects on specific vocalizations. Two separate studies showed that injection of amphetamine causes a selective increase in trills and a decrease in 50 kHz flat calls [3,53]. In sum, amphetamine induces trills and possibly other frequency-modulated calls. From this perspective, amphetamine elicits vocalizations very similar to those associated with social reward while food seems to elicit a non-overlapping set consisting of flat, step and other calls without trills. This suggests that vocalizations may be specific to certain types of reward, but more study is clearly needed.

We have previously provided evidence that certain categories of rat vocalizations are selectively emitted when rats are performing specific actions [4,16]. In the current study, we provide further evidence of the selective emission of calls with respect to behavior (Figure 6). The data are roughly consistent with findings from our previous study with the same strain of rats on the anticipatory period before play [16]. Admittedly, the plots are not identical because certain calls were omitted from each analysis due to low numbers, and the omitted calls were different in the two studies. Looking at the data from Day 7 in the Play Reward condition, we can see that walking is associated with trills, running with any trill call (trill, flat/trill combo and trill with jumps) and jumping is associated with a wide range of calls, most notably split, flat and downramp. In contrast, the data from Burke et al., [16], Figure 3C, largely agree for walk and partially agree for runs. In that study, running was associated with composite and trill with jumps, both of which have trills. The current associations between split calls and jumps agrees with Burke et al. [16] but that study included composite and multi-step, neither of which was common enough in the present dataset to analyze. Both studies agree that exploration and rearing are negatively correlated with vocalizations. Finally, it is interesting to note that the food-deprived groups show a marked lack of correlation between vocalizations and specific behaviors, especially on Day 7 (Figure 6, bottom two rows). Our data suggest that food deprivation not only reduces the number of vocalizations, but also may desynchronize their association with behaviors. We cannot rule out the possibility that this de-synchronization is due to low numbers of calls, but our shuffling method is robust and, if anything, tends to overestimate associations when vocal counts are low (which is why many rows and columns in that data are left blank). Taken together, this data strengthens the case that specific calls are tied to specific behaviors, but studies with a much higher number of vocalizations may be needed to iron out the specifics.

One limitation of our study is that it was restricted to male rats, as were the majority of studies cited above. Studies of the effects of sex differences suggest that, at least in juvenile play, sex differences in vocalizations are, if anything, subtle. Certainly, juvenile male rats play more than female rats [54,55]. It is hence not surprising that one recent study in Sprague-Dawley rats found lower overall vocalizations in juvenile females [47]. The authors also found decreases in specific calls (flat and step) but not others (trill). On the other hand, Gzielo et al. [46], also working with adolescent Sprague-Dawley rats, found no sex differences in either the number of vocalizations or their structure (duration, frequency and bandwidth). Similarly, we have recently compared vocalization sub-types in juvenile Long Evans rats during play and found no sex differences [45]. Adding a female group would certainly be valuable; however, as our study also used Long Evans rats, we have reasonable grounds to assume that their results would be substantively the same as those reported here.

## 5. Conclusions

In conclusion, our data add to the evidence that calls with trills are associated with social reward. It may not be possible to do a strict apples-to-apples comparison between food and social reward because the motivation for social interaction peaks in adolescence while food deprivation studies work better with adults. However, future studies that compare vocalizations to both forms of reward in rats of different ages but keeping all other parameters equivalent would be helpful to elicit exactly which calls are tied to social reward and which to food reward.

## Figures and Tables

**Figure 1 brainsci-11-01142-f001:**
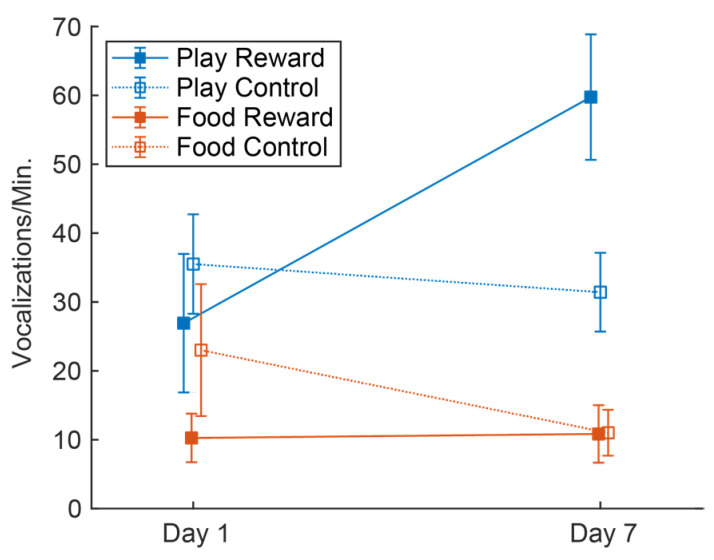
The average rate of 50 kHz vocalizations produced on day 1 and 7 of anticipatory testing. All error bars are standard error of the mean.

**Figure 2 brainsci-11-01142-f002:**
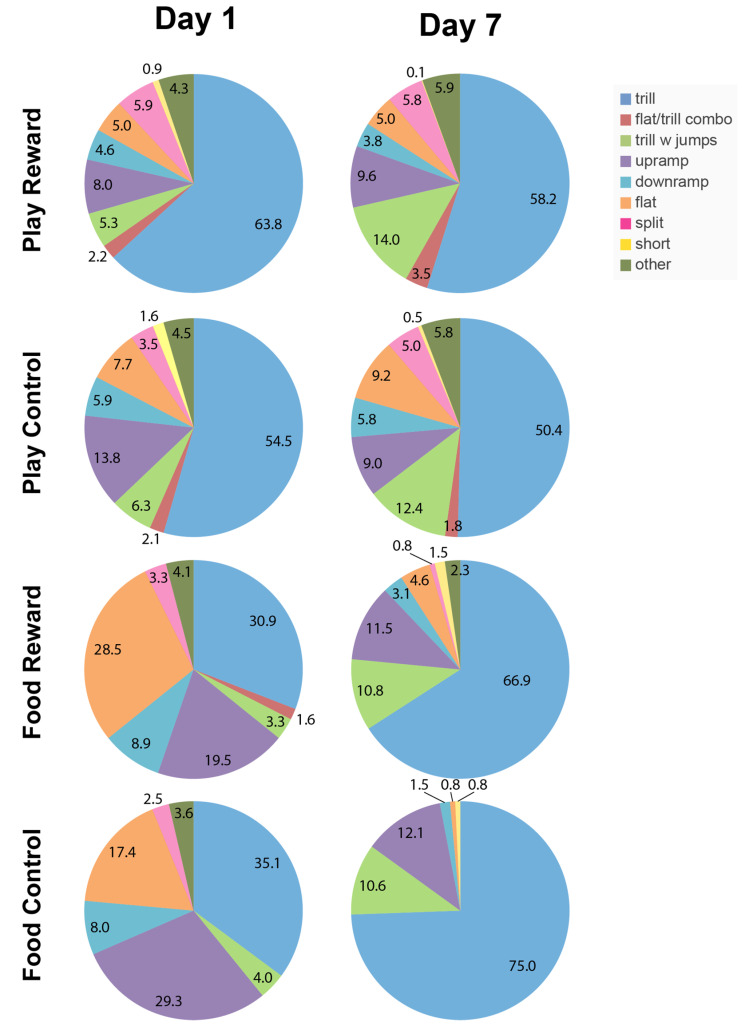
The average proportion of all commonly used call categories for each experimental group (Play Reward, Play Control, Food Reward, Food Control) on day 1 and 7 of testing are shown. Numbers show percentage of all calls for that group and testing day.

**Figure 3 brainsci-11-01142-f003:**
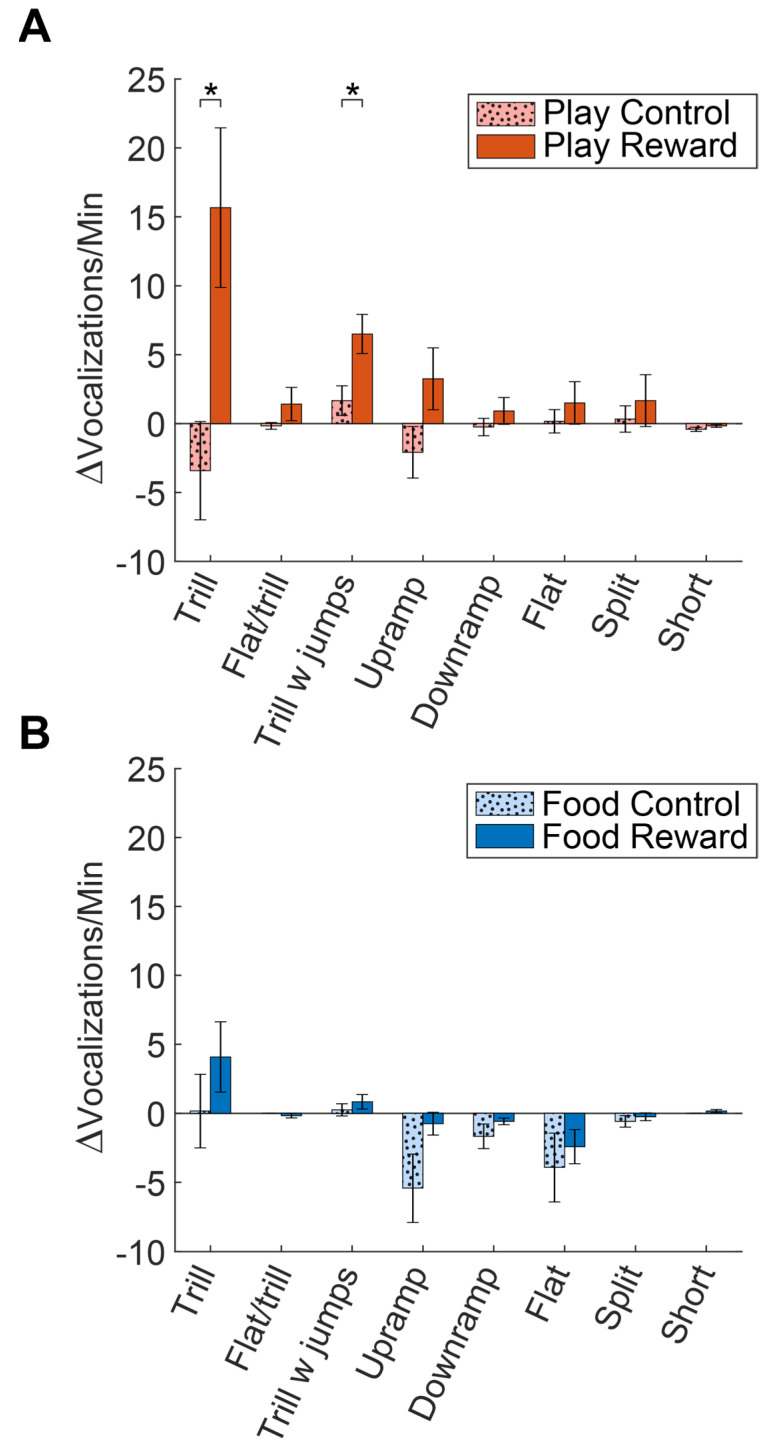
Comparison of the increase/decrease in call rates from day 1 to day 7 for each category of commonly produced calls. (**A**) Comparison of the change in call rates for Play Reward and Play Control groups. Asterisks denote comparisons that were statistically significant (*p* < 0.05). (**B**) Comparison of the change in call rates for Food Reward and Food Control groups.

**Figure 4 brainsci-11-01142-f004:**
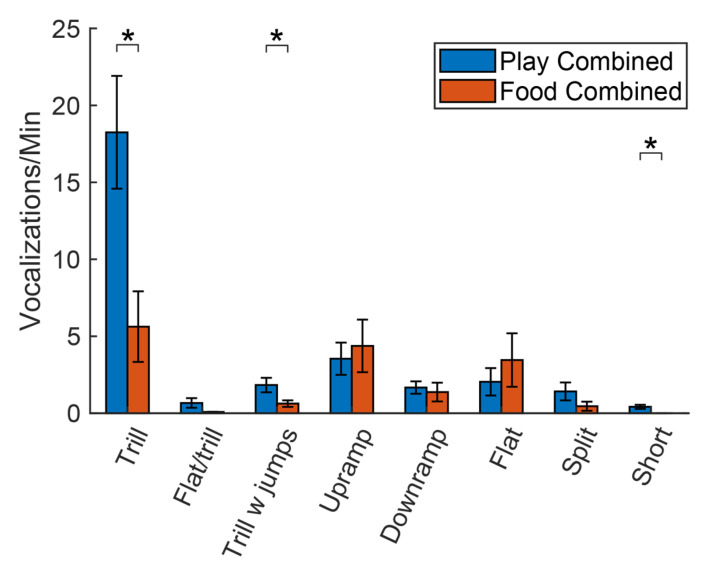
Comparison of the rates of calls on day 1 for all food and play groups. Play includes both Play Reward and Play Control while food includes both Food Reward and Food Control. Asterisks denote comparisons that were statistically significant (*p* < 0.05).

**Figure 5 brainsci-11-01142-f005:**
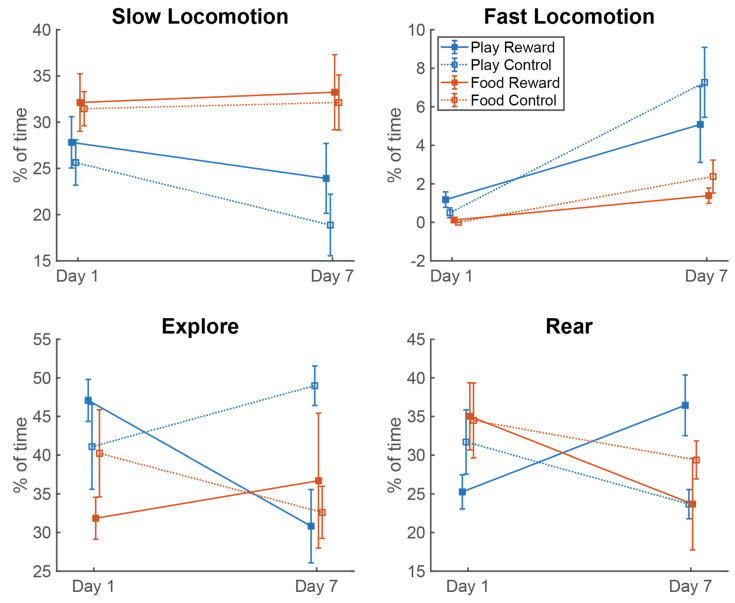
The average proportion of time spent in each of several key behaviors on day 1 and day 7 for each experimental group. Upper left shows the proportion of the 2 min test period spend in slow locomotion (single step, turn or walk). Upper right shows proportion of time spent in fast locomotion (running or jumping). Lower left shows time spent in exploratory behaviors. Lower right shows time spent rearing on hind legs. All error bars are standard error of the mean.

**Figure 6 brainsci-11-01142-f006:**
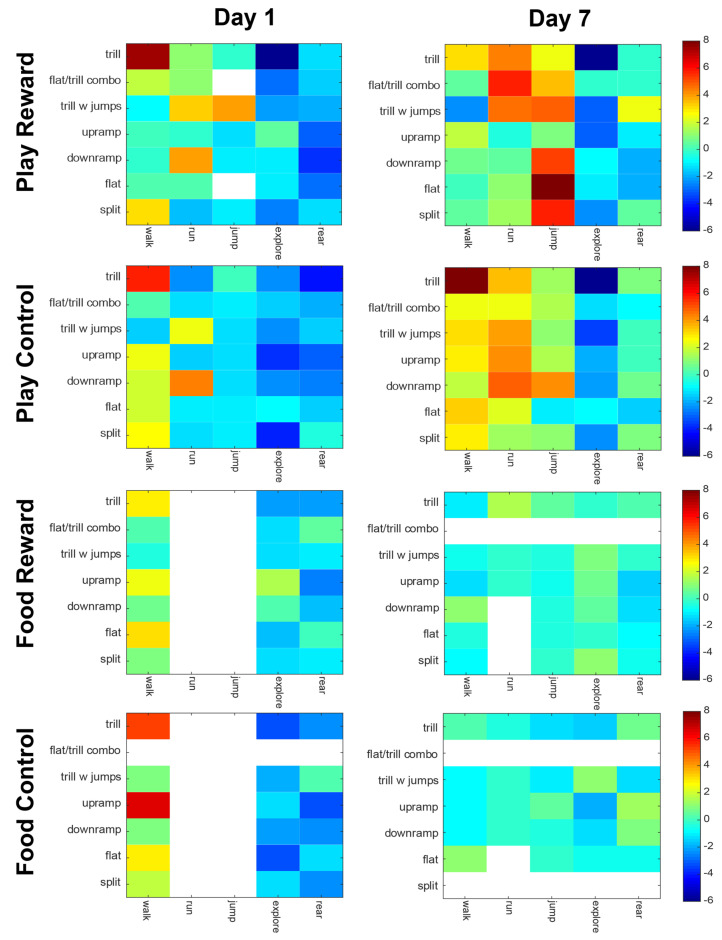
Association between types of calls and types of behavior are shown for the four groups on day 1 and 7 of the anticipation trials. Each matrix shows the strength of association, as a z-score, for each combination of behavior (x-axis) and vocalization category (y-axis). Deep red indicates the strongest positive association and deep blue the strongest negative association. The white sections indicate that either the behavior or the vocalization in that category did not have sufficient instances to run the analysis.

**Table 1 brainsci-11-01142-t001:** Description of the anticipatory behaviors that were scored.

Behavior	Description
Step	Removal of at least two paws from the ground in an alternating manner
Walk	Removal of all four limbs off the ground in an alternating manner (left paw and right hind limb move simultaneously followed by right paw and left hind limb) OR significant shift from one location to another (if all limbs are not visible)
Run	Only two limbs touch the ground at any given time; the rat may alternate two limbs at a time (as is seen during walking behavior) OR the rat may move two paws followed by two hind limbs at any given time; such movement is accompanied by the extension of the torso as the front limbs reach forward followed by flexion of the torso as the hind limbs are removed from the ground and placed under the body
Jump	Up jump: the two front limbs leave the ground followed by the hind limbs while body is lifted into the air, then all limbs touch the ground simultaneously or closely one after the otherForward jump: the two front limbs are extended forward and removed from the ground followed by the removal of the hind limbs from the ground; this behavior is accompanied by the extension of the torso as the front limbs reach forward followed by flexion of the torso as the hind limbs are removed from the ground
Turn	Turn with one or both front limbs at a 45-, 90- or 180-degree angle OR turn with three or more limbs at a 360-degree angle. Turning may also be preceded by a stepping or walking pattern or followed by a rear (see below for the operational definition of rearing behavior)
Explore	Immobile; may extend one front limb; turning of head so as to examine the surrounding area
Dig	Vigorous forward and backward motion of front limbs while significantly displacing bedding
Rear	Standing on rear limbs with both front paws off ground (either free standing or against wall)
Shake	Vigorous side-to-side shudder of head, neck and trunk
Groom	Licking of paws; wipes/rubs face and nose; wipes behind ears, neck and/or downward to either side of the body; may grab fur and nibble with teeth. Grooming may consist of a variation of these behaviors many consecutive times. However, grooming is typically initiated by wiping of the nose or face and followed by grooming of the neck and body
Scratch	Rapid movement of hind limb with the claws rubbing against head, neck or side
Rest	Immobile; may turn head, but significantly less than is seen during exploration

## Data Availability

Data available upon request.

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
