# Peer review of "Rat 50 kHz Trill Calls Are Tied to the Expectation of Social Interaction"

_brainsci, 2021, doi:10.3390/brainsci11091142_

Round 1

Reviewer 1 Report

This study evaluated the subtype of rat USVs associated with anticipation of food or social rewards. The results indicated that ‘trill’ USVs were associated with anticipation of same-sex social encounters.

Overall impression:

Overall, this article is well-written, clear, and concise. The study is well designed, and the methods are clearly described and address the research question. The results are clearly presented and easily interpretable, although more statistical information should be provided. The discussion clearly summarizes the results and compares the results to relevant literature with appropriate interpretation. While the study intricately analyzes USV subtype and behaviors (and their correlations) in two appetitive situations, because it is not clear in the introduction how this article builds on the existing literature, it is difficult to determine the novel contributions of this article. Therefore, this article would benefit from major revisions in the introduction by integrating the current study with some highly-related preliminary investigations, detailed below. Further, this study reports male-only vocalizations and this needs to be contextualized. Finally, raven pro was used tally the USV subtypes but no USV acoustic measures were reported (e.g., duration, peak frequency, etc.), which is an unfortunate (but acceptable limitation) of the study. Acoustic data may have shed some light on results as they could be compared with other studies and this should be discuseed. Overall, this is very solid work.

Major weaknesses:

  1. Introduction: While the introduction is clear and provides background information regarding USV subtypes, the following relevant articles should be included in the literature review. Particularly, the Mulvihill article which is highly related. Right now, the novel contribution of this study is missing from the introduction.
  • Burgdorf, Jeffrey, et al. "Ultrasonic vocalizations of rats (Rattus norvegicus) during mating, play, and aggression: Behavioral concomitants, relationship to reward, and self-administration of playback.." Journal of comparative psychology4 (2008): 357.
  • Burke, C. J., Kisko, T. M., Swiftwolfe, H., Pellis, S. M., & Euston, D. R. (2017). Specific 50-kHz vocalizations are tightly linked to particular types of behavior in juvenile rats anticipating play. PLoS One, 12(5), e0175841.
  • Mulvihill, K. G., & Brudzynski, S. M. (2018). Non-pharmacological induction of rat 50 kHz ultrasonic vocalization: Social and non-social contexts differentially induce 50 kHz call subtypes. Physiology & behavior, 196, 200-207.
  • Opiol, H., Pavlovski, I., Michalik, M., & Mistlberger, R. E. (2015). Ultrasonic vocalizations in rats anticipating circadian feeding schedules. Behavioural brain research284, 42-50.
  • Takahashi, N., Kashino, M., & Hironaka, N. (2010). Structure of rat ultrasonic vocalizations and its relevance to behavior. PloS one5(11), e14115.
  • Willey, A. R., & Spear, L. P. (2012). Development of anticipatory 50 kHz USV production to a social stimuli in adolescent and adult male Sprague-Dawley rats. Behavioural brain research, 226(2), 613-618.
  • Wöhr, M., Houx, B., Schwarting, R. K., & Spruijt, B. (2008). Effects of experience and context on 50-kHz vocalizations in rats. Physiology & behavior93(4-5), 766-776.

Methods:

  1. The statistics used for figures 2-5 were not outlined in the methods, only the statistics for figure 6 were planned. Please provide these details and clarify. Perhaps a table outlining the USV and behavior dependent variables and statistical comparisons would be helpful.
  2. Line 180. Please report inter and intra-rater reliability for USV categorization.

Discussion:

  1. Male rats were used in this study which limits its interpretation to one sex (male) and this should be stated throughout the manuscript-when comparing literature, etc. Please ensure this is clear throughout the manuscript.
  2. The term ‘lonely’ (although clever and concise) anthropomorphizes rat behavior and should be adjusted.
  3. Raven pro was used tally the USV subtypes but no USV acoustic measures were reported (e.g., duration, peak frequency, etc), which is an unfortunate limitation of the study and needs to be discussed. USV acoustic measures would have been helpful in interpreting results that differed from other research and providing acoustics in the current study could then be interpreted by other investigators with relevance to the broader context of this type of research. For this paper, this is noted as an acceptable limitation.

Minor weaknesses

  1. Line 30. Not all 22-kHz USVs are long and unmodulated. Please reword to clarify.
  2. Line 39. Please include a reference for the VTA study.
  3. Replace ‘gave’ USVs with ‘produced.’
  4. Line 108. While the major of USVs are produced below 100-kHz, harmonics of USVs can reach well above 100-kHz. The authors should consider using a microphone that will capture the entire USV frequency range and discuss this as a limitation.
  5. Line 129. By stating that the play was in complete darkness, it is unclear if the rats were on a reversed light cycle during the experiment. If the rats were on a reversed light cycle, it should be stated. If not, the authors should list this as a limitation to the study.
  6. Line 145-146. In the future the authors should consider weight restriction for each individual animal’s baseline weight, not in relation to a cage mate.
  7. Line 191. 200ms of error seems to be a long duration in the context of USVs. Why was 200ms chosen?
  8. Figure 2. While the pie charts are visually appealing, this data should be presented numerically.
  9. Line 262. ‘Separated’ instead of ‘separately.’
  10. Paragraph 3-6 of the Discussion. These references should be discussed in the introduction.
  11. 472-473. This statement is a bit over-reaching. The lack of correlation may be influenced by the low number of calls. The next sentence does temper this statement somewhat, but consider rephrasing to modify this statement further.

Reviewer 2 Report

The paper is not particularly groundbreaking, but it presents a solid work with an interesting oucome.

My only suggestion is that authors could try to qunatify the call type diversity, for instance with entropy or Gini coefficient, which could bring some further insignt over simple frequency comparisons.

Reviewer 3 Report

In the manuscript by Burke, Markovina, Pellis, and Euston, the authors investigate the association of the 50kHz vocalizations in juvenile male rats to two types of attractive rewards (play and food) under the contexts of social isolation and food deprivation, respectively.  While the calls rats emit and the reward system is well-known, the effect of specific reward types remains unclear.  The authors utilized a series of behavioral tests to identify that more calls are produced when anticipating the play rewards than the food rewards. The authors then perform a series of association tests between types of calls and types of behaviors on two experimental points: Day 1 vs. Day 7. This allowed analysis of the detailed behavioral status of different categories of call emission. The data suggest that, in juvenile rats, calls with trills are associated with social reward (play), as the motivation for social interaction peaks in adolescence.

The experiment methods chosen are adequate. The associations of behavior and call categories are excellent, showing that the same call type may emitted together with different activity over the time. I have a few concerns about the comprehensiveness of the study. The increase of vocalization under social isolation only evidently showed in the comparison between Day 1 and Day 7. No data was illustrated whether the rats emitted more calls during the Day 2 to Day 6. As such, the limited number of subjects/sample size may not hold up the interpretations, if more rats were tested, or if prolonged experiments (e.g. Day 14) might alter the findings. 

It would be helpful if the authors could reorganize the introduction. Some paragraphs lacked focus and the relevant background information was scattered over different paragraphs. For example, the association of 50kHz calls of rats and sexual related interactions were mentioned in L45-47, in L56-58, and again in L61-64.  I suggest sorting the introduction from high to low levels of rewarding: the drug/electric induced 50kHz calling, sexual motivation (please include the 50kHz emitting during pre-intromission and pre-ejaculation period, both copulatory acts are highly rewarding, see Barfield et al. 1979), food, and then social contact (conspecific play, and heterospecific interaction). As the high rewarding activity (addictive drugs or sex) associated vocalizations have been intensively studied, authors may emphasis whether rats emit calls selectively in relatively low rewarding contexts.

Primary rewards are a class of rewarding stimuli which facilitate the survival of oneself and offspring (food, sex, ect.). It is expected to see rats perform more calls to food than to social contact. The play rewards appear to have a high impact in ultrasonic calls. The authors discussed the possible explanation of why more calls are produced prior to social interaction than primary reward (food, in this study) in young rats. How about the comparison between play and another primary rewards (e.g. sex), some discussion for this is warranted. 

Minor suggestion:

Please revise the sentence in L78-80.

Fig. 2 Please define the mean frequency or mean proportion of each call type in the pie chart and examples of categorized 50-kHz calls.

How similar are the calls distributed during the 2 min recording in Day 1 and Day 7? It might be helpful to indicate the intervocalization interval. 

Does more USV calling  indicate a higher level of reward?
